# Isoespintanol Antifungal Activity Involves Mitochondrial Dysfunction, Inhibition of Biofilm Formation, and Damage to Cell Wall Integrity in *Candida tropicalis*

**DOI:** 10.3390/ijms241210187

**Published:** 2023-06-15

**Authors:** Orfa Inés Contreras Martínez, Alberto Angulo Ortíz, Gilmar Santafé Patiño, Ana Peñata-Taborda, Ricardo Berrio Soto

**Affiliations:** 1Biology Department, Faculty of Basic Sciences, Universidad de Córdoba, Montería 230002, Colombia; 2Chemistry Department, Faculty of Basic Sciences, Universidad de Córdoba, Montería 230002, Colombia; aaangulo@correo.unicordoba.edu.co (A.A.O.); gsantafe@correo.unicordoba.edu.co (G.S.P.); 3Biomedical and Molecular Biology Research Group, Universidad del Sinú E.B.Z., Montería 230001, Colombia

**Keywords:** antifungal mechanism, isoespintanol, *Candida tropicalis*, antibiofilm activity

## Abstract

The growing increase in infections caused by *C. tropicalis*, associated with its drug resistance and consequent high mortality, especially in immunosuppressed people, today generates a serious global public health problem. In the search for new potential drug candidates that can be used as treatments or adjuvants in the control of infections by these pathogenic yeasts, the objective of this research was to evaluate the action of isoespintanol (ISO) against the formation of fungal biofilms, the mitochondrial membrane potential (ΔΨm), and its effect on the integrity of the cell wall. We report the ability of ISO to inhibit the formation of biofilms by up to 89.35%, in all cases higher than the values expressed by amphotericin B (AFB). Flow cytometric experiments using rhodamine 123 (Rh123) showed the ability of ISO to cause mitochondrial dysfunction in these cells. Likewise, experiments using calcofluor white (CFW) and analyzed by flow cytometry showed the ability of ISO to affect the integrity of the cell wall by stimulating chitin synthesis; these changes in the integrity of the wall were also observed through transmission electron microscopy (TEM). These mechanisms are involved in the antifungal action of this monoterpene.

## 1. Introduction

*Candida tropicalis* is one of the most important non-albicans candida (NAC) species, due to its high incidence of systemic candidiasis and greater resistance to commonly used antifungals [1]. This yeast has been considered the second-most virulent *Candida* species, only preceded by *C. albicans*. It expresses a wide range of virulence factors, including adhesion to buccal epithelium and endothelial cells, the secretion of lytic enzymes, hyphal budding, and the phenomenon called phenotypic switching, which allows them to rapidly adapt in response to environmental challenges. This yeast has been recognized as a strong producer of biofilms, surpassing *C. albicans* in most studies [2]. *Candida tropicalis* is an opportunistic pathogen that affects immunosuppressed individuals and is capable of spreading to vital organs [3]. It has been reported that this yeast is associated with higher mortality compared to *C. albicans* and other NAC species, showing a greater potential for dissemination in neutropenic individuals; it is associated with malignancy, cancer patients, patients with long-term catheter use, and broad-spectrum antibiotic therapy [4]. In Colombia, candidemia is a frequent cause of infection in the bloodstream, especially in intensive care units (ICU), representing 88% of fungal infections in hospitalized patients, with a mortality rate between 36% and 78%. The incidence in Colombia is superior to that reported in developed countries and even in other Latin American countries [5].

In this context, the search for compounds with antifungal potential against these pathogens is urgent. Natural products feature prominently in the discovery and development of many drugs used today with recognized medicinal properties; especially, plants have played a major role as a source of specialized metabolites with curative effects, which can be used directly as bioactive compounds, as drug prototypes, and/or as pharmacological tools for different targets [6].

Isoespintanol (ISO) (2-isopropyl-3,6-dimethoxy-5-methylphenol) has been recognized as a natural bioactive compound. It is a monoterpene that was first obtained from *Eupatorium saltense* (Asteraceae) [7], whose synthesis has been reported [8], and also from *Oxandra xylopioides* (Annonaceae) [9]. The biological potential of this compound as a natural antioxidant [10], anti-inflammatory [9], antispasmodic [11], vasodilator [12], cardioprotective [13], and cryoprotectant in canine semen [14], as well as its insecticidal activity [15] and antifungal activity against phytopathogens of the genus *Colletotrichum* [16], has been documented. We have also reported its potential against human pathogens, specifically intra-hospital bacteria [17] and yeasts of the genus *Candida*, reporting its action against the cell membrane, its ability to induce intracellular reactive oxygen species, and its ability to eradicate mature biofilms as responsible for its antifungal activity [18,19]. Continuing with the study of this compound, we hypothesize that ISO may have other targets of action against *C. tropicalis*. The purpose of this research was to evaluate other target sites of action of ISO by investigating its action against the mitochondrial membrane potential (ΔΨm), its ability to prevent the formation of fungal biofilms, and its effect on the integrity of the cell wall, thus contributing to the knowledge of the mechanisms of action of this monoterpene, which could serve as adjuvants in the treatment and control of these pathogenic yeasts with resistance to conventional antifungals.

## 2. Results

### 2.1. Obtaining and Identification of Isoespintanol

ISO was obtained as a crystalline amorphous solid from the petroleum benzine extract of *O. xylopioides* leaves, and its structural identification was performed by GC-MS, ^1^H-NMR, ^13^C-NMR, DEPT, COSY ^1^H-^1^H, HMQC, and HMBC. Information related to obtaining and identifying the ISO was reported in our previous study [19].

### 2.2. Antifungal Susceptibility Testing

Table 1 shows the minimum inhibitory concentration (MIC_90_ and MIC_50_) and minimum fungicidal concentration (MFC) values of the ISO against the clinical isolates of *C. tropicalis* evaluated in this study and in previous work [19].

### 2.3. Effect of ISO on the Formation of Biofilms

ISO inhibited the formation of biofilms in all *C. tropicalis* isolates evaluated, as shown in Figure 1a. Biofilm formation was significantly less in ISO-treated cells compared to untreated cells used as a negative control and AFB-treated cells. As observed in Figure 1b, ISO showed higher percentages of inhibition of biofilm formation compared to AFB. Table 2 presents the percentages of inhibition of biofilm formation for each isolate of *C. tropicalis*.

### 2.4. Effect of ISO on Mitochondrial Membrane Potential (ΔΨm)

Our results show that the mitochondrial function of *C. tropicalis* was significantly affected after ISO exposure. In Figure 2, the ΔΨm loss of ISO-treated cells compared to untreated cells and H_2_O_2_-treated cells is observed. Rhodamine 123 (Rh123) accumulates in the highly negatively charged interior of mitochondria, and its fluorescence intensity reflects the ΔΨm across the inner mitochondrial membrane. A loss of ΔΨm results in leakage of Rh123 from mitochondria, with a consequent decrease in fluorescence.

As shown in flow cytometric analysis (Figure 3), fluorescence intensity was significantly decreased in ISO-treated cells, suggesting that ISO caused mitochondrial depolarization of *C. tropicalis*, causing mitochondrial dysfunction and consequent death.

### 2.5. Effect of ISO on Cell Wall Integrity

All ISO-treated *C. tropicalis* isolates showed higher chitin content than untreated control cells, as revealed by staining index (SI) values. However, cells treated with caspofungin (CASP) showed the highest chitin concentration compared to the other two experimental groups. Figure 4 represents the SI values calculated for the three experimental groups.

Figure 5 shows the fluorescence emitted by cells treated with ISO, cells without treatment (INO), and cells treated with CASP after staining with 2.5 µg/mL calcofluor white (CFW). Histograms show higher CFW fluorescence in ISO-treated cells (i.e., higher chitin content) compared to untreated cells and the highest CFW fluorescence was revealed in CASP-treated cells.

#### Transmission Electron Microscopy (TEM)

TEM showed damage to the cell wall integrity and morphology of ISO-treated *C. tropicalis* isolates compared to untreated control cells. As shown in Figure 6a,b, untreated cells show intact cell morphology with intact cell walls. However, after ISO treatment, as observed in Figure 6c–e, the cells revealed damage to the integrity of their envelope, with partially dissolved cell walls.

### 2.6. Isoespintanol Cytotoxicity Assays

Cytotoxicity assays showed different profiles in the cytotoxicity methods tested, revealing statistically significant differences (*p* < 0.05) between cells treated with ISO and untreated cells used as negative controls. As shown in Figure 7a,b, cytotoxicity was dose dependent. The higher the ISO concentration, the lower the percentage of cell viability observed. The inhibitory concentration 50 (IC_50_) obtained by the crystal violet (CV) assay was 48.64 µg/mL, significantly lower (*p* < 0.0001) than that found by the 3-(4,5-dimethylthiazol-2-yl)-2,5-diphenyl tetrazolium bromide (MTT) (77.34 µg/mL).

Figure 8 shows the dose–response curves of the viability profile of VERO cells treated with ISO, using the MTT and CV assays. IC_50_ values were calculated from the fit (R^2^ > 0.95) of the Hill slope curve using nonlinear regression analysis in GraphPad Prism Software version 8.0.

## 3. Discussion

The increase in *C. tropicalis* infections in recent years and the resistance to commonly used antifungals expressed by these yeasts, especially in immunocompromised patients, have made these candidemias a great challenge, not only due to the increase in rates of morbidity and mortality but also due to the financial costs at a global level. Therefore, the search for new effective and safe compounds with antifungal potential is urgent today.

In this research, we demonstrated that the ISO monoterpene extracted from *O. xylopioides* inhibits the formation of fungal biofilms. Consistent with our previous work, where we reported its ability to inhibit mature biofilms in *C. tropicalis* [19], with inhibition percentages between 20.3% and 25.8% higher than the percentages shown with AFB (7.2% and 12.4%). Moreover, the effect of ISO on the mature biofilms of other *Candida* species has been reported, even in those isolates where AFB did not show an effect [18]. Our results show percentages of inhibition in the formation of biofilms by ISO greater than 50% (between 59.18% and 89.35), higher than those shown by AFB. This information is of great value considering that *C. tropicalis* is a pathogen well known for the formation of strong biofilms as a result of its high metabolic activity [20]. These biofilms represent one of the main virulence factors of these yeasts and vary depending on the origin of the infection [21]. They have been associated with the high mortality caused by these pathogens, probably due to the low permeability of the matrix to commonly used antifungals [22]. Comparing the efficacy of ISO against AFB, we highlight the role of ISO with percentages higher than 80% in the inhibition of biofilm formation, higher than those shown by AFB. Studies reported by [20] have documented the ability of liposomal AFB to inhibit the growth of biofilms but its ineffectiveness in eradicating mature biofilms, even at high doses. These results allow us to suggest that ISO could be a promising alternative to combat multiresistant pathogenic yeasts that form biofilms [23].

On the other hand, taking into account that many antifungal agents of vegetable origin can inhibit the function of the fungal mitochondrial electron transport chain, leading to the reduction of ΔΨm [24], which is an important indicator of mitochondrial function [25,26,27], we investigated the possible effect of ISO against *C. tropicalis* mitochondria. In this work, we use Rh123, a permeable lipophilic cationic fluorochrome [28], which selectively accumulates in the mitochondria of active cells. This specific interaction depends on the high transmembrane potential maintained by functional mitochondria; therefore, the dissipation of ΔΨm by ionophores or electron transport inhibitors eliminates the selective mitochondrial association of these compounds [29]. Consequently, mitochondrial activation induces quenching of Rh123 fluorescence, and the rate of fluorescence decay is proportional to ΔΨm [30,31]. The loss of ΔΨm is considered the earliest event in the apoptotic cascade, where the mitochondrial permeability transition pore (MPTP) opens and leads to the collapse of ΔΨm and irreversibly initiates cell apoptosis [32,33,34]. Therefore, mitochondrial depolarization is an indicator of mitochondrial-mediated apoptosis. We reported the loss in ΔΨm in yeasts treated with ISO as being significantly higher compared to the control group (untreated cells) and cells treated with H_2_O_2_. These results allow us to infer that ISO causes cell apoptosis mediated by mitochondria in *C. tropicalis*, this being another mechanism responsible for the antifungal activity of this monoterpene.

We also evaluated the effect of ISO on the integrity of the cell wall through the measurement of chitin content based on CFW staining and analysis by flow cytometry. Chitin is one of the main structural components in fungal cell walls. Together with β-1,3-glucan, they play a fundamental role in maintaining the integrity of the cell wall, giving it structural rigidity during growth and morphogenesis [35,36]. All ISO-treated isolates showed higher chitin content (revealed by higher CFW fluorescence intensity) compared to untreated strains. These results are consistent with studies reported by other researchers [37,38,39,40], who postulate that perturbation of cell wall synthesis in some yeasts, either by mutations in synthesis-related genes or by adding compounds that interfere with normal cell wall assembly, triggers a compensatory response to ensure the integrity of the cell wall; this response includes increased levels of chitin in the cell wall, suggesting that cell wall stress in fungi can generally lead to activation of the chitin biosynthetic pathway. This allows us to suggest that ISO could be acting on the integrity of the cell wall of these yeasts and inducing the compensatory synthesis of chitin. Reported studies [39] show the ability of these yeasts to grow in the presence of CASP, an antifungal that acts on the synthesis of β-1,3-glucan. The action of CASP on these yeasts causes them to activate a compensatory pathway, inducing the synthesis of chitin. This is consistent with our results, which reveal a higher chitin content in CASP-treated cells compared to untreated and ISO-treated cells.

Finally, we evaluated the cytotoxic effect of ISO on VERO cells through the MTT and CV methods. The results showed significant differences between the cytotoxicity methods used. The IC_50_ obtained by the CV assay was significantly lower (48.64 µg/mL) than that found by the MTT assay (77.34 µg/mL). This observation can be explained by the nature of each test; the MTT assay is mainly based on the enzymatic conversion of MTT in mitochondria, so it could be influenced by inhibitors of mitochondrial components [41]. Therefore, a cytotoxicity assay based on mitochondrial respiratory activity would give early signs of toxicity after exposure to mitochondrial toxicants; ISO affects mitochondrial function, which could influence the cytotoxicity results of this method. Furthermore, the MTT assay can be significantly influenced by compounds that modify cell metabolism and reaction conditions [42]. Since various investigations have linked mitochondrial metabolism to the progression of cancer and other pathologies [43], ISO could be used in a tumor cell model to evaluate its effect on them. Furthermore, in tumor cells, there are higher levels of reactive oxygen species than in their non-tumor cells of origin, and, therefore, they must employ various metabolic strategies to prevent oxidative stress [44]. Taking into account the effect of ISO on the induction of intracellular reactive oxygen species [19], this could help in its action against these cells. On the other hand, with the CV method, cells that undergo cell death lose their adherence and are subsequently lost from the cell population, which reduces the amount of CV staining in the wells. Therefore, the amount of dye absorbed depends on the total DNA and/or protein content in the culture, thus allowing estimation of the number of viable cells in the wells [45]. It has been previously reported that different cytotoxicity assays may give different results depending on the test agent used and the cytotoxicity assay employed [46]. For this reason, it is important to consider what effect is expected, that is, the mechanism of action of the agent evaluated.

Our results are consistent with other studies that show that ISO at low concentrations does not have toxic effects. ISO cytotoxicity assays on human peripheral blood lymphocytes have indicated that at 3.0 μM, 8.0 μM, and 80 μM, it has no genotoxic or cytotoxic effects on these cells, and at concentrations between 3 and 1620 μM, it shows a protective effect on damage to the DNA from lymphocytes induced by H_2_O_2_, suggesting that at low concentrations it can be used without expecting negative effects on human health [47]. Likewise, the cytotoxic effect of ISO against murine macrophages (RAW 264.7) has been investigated, revealing that ISO at 100 µM does not have significant cytotoxic effects against these cells, considering the possible use of ISO as a food additive [48]. The findings on the biological potential of ISO could serve as a starting point to understand the mechanisms of action at low doses. Previous research has established that mitochondria are a fundamental element of apoptotic signaling [44], so the cell death mechanisms observed in this study support the effects on the indicated targets.

Likewise, the cytotoxicity results could have implications in the search for new therapeutic alternatives and the reduction of the adverse effects of fungal diseases. This evidence gives rise to exploring the individual or combined effects of ISO and its mechanisms of action on cell proliferation and its association with mitochondrial metabolic pathways. The integration of these approaches in future research is required to contribute to the understanding of its antifungal activity and the potential use of this compound in the clinical setting.

## 4. Materials and Methods

### 4.1. Reagents

RPMI 1640, phosphate-buffered saline (PBS), and yeast peptone dextrose broth (YPD) were obtained from Thermo Fisher Scientific, Waltham, MA, USA; 3-N-morpholinopropanesulfonic acid (MOPS) was obtained from Merck; potato dextrose broth (PDB), sabouraud dextrose agar (SDA), sabouraud dextrose broth (SDB), amphotericin B (AFB), rhodamine 123 (Rh123), calcofluor white (CFW), caspofungin (CASP), and crystal violet (CV) used in this study were obtained from Sigma-Aldrich, USA; glacial acetic acid was obtained from Carlo Erba Reagents, Italy.

### 4.2. Strains

Seven clinical isolates of *C. tropicalis* (001 to 007) were used in this study. The isolates were cultured from blood cultures and tracheal aspirate samples from hospitalized patients at the Salud Social S.A.S. from the city of Sincelejo, Colombia. All microorganisms were identified by standard methods: Vitek 2 Compact, Biomerieux SA, YST Vitek 2 Card, and AST-YS08 Vitek 2 Card (Ref 420739). SDA medium and BBL CHROMagar Candida medium were used to maintain the cultures until the tests were carried out. The identification of one of the *C. tropicalis* isolates was confirmed through a genome-wide taxonomic study (information reported in previous work) [19].

### 4.3. Antifungal Susceptibility Testing

The ISO minimal inhibitory concentration (MIC) against *C. tropicalis* was defined as the lowest concentration at which 90% (MIC_90_) of fungal growth was inhibited, compared to the negative control (untreated cells). MICs were established following the protocols described in the Clinical Laboratory Standards Institute (CLSI) method (M27-A3) [49] and The European Committee for Antimicrobial Susceptibility Testing (EUCAST) [50]. The MIC_90_, MIC_50_ (lowest concentration at which 50% of fungal growth was inhibited), and MFC (minimum fungicidal concentration) of ISO against *C. tropicalis* were previously reported [19].

### 4.4. Effect of ISO on the Formation of Biofilms

The effect of ISO on biofilm formation was evaluated following the protocol described in previous works [19]. In the present study, the ISO MIC was added at the time of inoculation with *C. tropicalis*. The yeast colonies in SDA were used to standardize the inoculum until it reached a concentration of 10^6^ cells/mL. Then, in 96-well boxes, 200 µL of the inoculum in YPD broth with the ISO MIC for each isolate was cultured in each reaction well and incubated at 37 °C for 48 h. Subsequently, the broth was removed from the microplates, and the biofilms in the wells were washed three times with deionized water. Three replicates of each sample were made. Cultures without ISO were used as a negative control, and AFB (4 µg/mL) was used as a positive control. The percentage reduction in biofilm formation was quantified by staining the wells with 0.1% crystal violet for 20 min. The wells were washed with deionized water until excess dye was removed. Finally, the samples were treated with 250 µL of glacial acetic acid, and the absorbance values were measured at 590 nm (OD_590_) using a SYNERGY LX (Biotek, Wichita, KS, USA) plate reader. Biofilm production was grouped into the following categories: OD_590_ < 0.1: non-producers (NP), OD_590_ 0.1–1.0: weak producers (WP), OD_590_ 1.1–3.0: moderate producers (MP), and OD_590_ > 3.0: strong producers (SP). Biofilm reduction was calculated using the following equation: % reduction in biofilm formation: AbsCO − AbsISO/AbsCO × 100
where AbsCO: absorbance of the control and AbsISO: absorbance of the sample treated with ISO.

### 4.5. Effect of ISO on Mitochondrial Membrane Potential (ΔΨm)

To evaluate the effect of ISO on ΔΨm, yeasts were stained with Rhodamine 123 (Rh123) as described by Chang [25] with minor modifications. Fungal cells (3 × 10^8^ UFC/mL) were treated with the ISO MIC for 1 h, harvested by centrifugation, resuspended with 25 µM Rh123 (in 50 mM sodium citrate), and incubated at 30 °C for 10 min. After staining, cells were washed three times with PBS, and the fluorescence intensity was measured using the BD FACS CANTO II flow cytometer and analyzed with the BD FACS DIVA software version 6.1.3. (Ext: 488 nm/Emi:525 nm). Cells without ISO treatment were used as negative controls, whereas cells treated with 15 mM hydrogen peroxide (H_2_O_2_) for 1 h were used as positive controls.

Rh123 is a permeable lipophilic cationic fluorochrome [28], and it selectively accumulates in the mitochondria of active cells. This specific interaction depends on the high transmembrane potential maintained by functional mitochondria; therefore, the dissipation of the mitochondrial transmembrane potential by ionophores or electron transport inhibitors eliminates the selective mitochondrial association of these compounds [29]. Consequently, mitochondrial activation induces quenching of Rh123 fluorescence, and the rate of fluorescence decay is proportional to ΔΨm [30,31].

### 4.6. Effect of ISO on Cell Wall Integrity

Damage to the integrity of the fungal wall by ISO was evaluated by measuring the chitin content of the cell wall using CFW staining, following the protocol described in [39] with minor modifications. CFW is a water-soluble fluorescent dye that exhibits selective binding to fungal cell walls (specific for chitin) [51] and fluoresces blue/green when illuminated with UV light [52]. Yeasts grown in YPD broth (1 × 10^6^ cells/mL) at 35 °C were treated with ISO MIC for 2 h and stained with CFW [2.5 µg/mL] for 15 min in the dark. Subsequently, cells were washed and resuspended in PBS and finally analyzed on a BD FACS CANTO II flow cytometer (pacific blue channel: 405–450/50 nm; 20,000 events per assay) using the BD FACS DIVA software version 6.1.3. All experiments were performed in triplicate. A staining index (SI) was defined [39], whose value was directly related to the amount of chitin and took into account the different levels of autofluorescence. The mean fluorescence intensity (MFI) emitted from stained (positive population) and unstained (negative population) yeasts was analyzed, and in each experiment, the SI was calculated using the following equation:SI: (MFIpp − MFIpn)/2 × SDpn
where MFIpp: mean fluorescence intensity of the positive population; MFIpn: mean fluorescence intensity of the negative population; and SDpn: standard deviation of the negative population.

#### Transmission Electron Microscopy (TEM)

The damage to the integrity of the cell wall as well as the general morphology of *C. tropicalis* after treatment with ISO was also analyzed through TEM, following the protocol described in previous studies [19]. The concentration of *C. tropicalis* was adjusted to 10^6^ CFU/mL; the suspension was mixed with ISO (200 µg/mL) and incubated at 37 °C for 24 h. Subsequently, the cells were collected and fixed in 2.5% glutaraldehyde in phosphate buffer pH 7.2 at 4 °C; they were centrifuged at 13,000 rpm for 3 min, and the button at the bottom of the vial was postfixed in 1% osmium tetroxide in water for 2 h at 4 °C. Then, pre-imbibition with 3% uranyl acetate was performed for 1 h at room temperature, after which the cells were dehydrated in an ethanol gradient (50% for 10 min, 70% for 10 min, 90% for 10 min, 100% for 10 min), acetone:ethanol (1:1) for 15 min, and embedded in SPURR epoxy resin. The samples were cut in a Leica EM UC7 ultramicrotome at 130 nm thickness, contrasted with 6% uranyl acetate and lead citrate, and then finally observed in a JEOL 1400 plus transmission electron microscope (JEOL Ltd., Tokyo, Japan). The photographs were obtained with a Gatan Orius CCD camera (Gatan Inc., Pleasanton, CA, USA).

### 4.7. Isoespintanol Cytotoxicity Assays

The cytotoxicity assays were carried out using immortalized epithelial cells from the African green monkey kidney (*Cercopithecus aethiops*) (VERO), which, due to their homology with human cells and their easy culture, are commonly used as a useful model to evaluate in vitro the cytotoxic activity of natural products [53,54]. These assays were performed using the crystal violet (CV) and 3-(4,5-dimethylthiazol-2-yl)-2,5-diphenyltetrazolium bromide) (MTT) methods.

#### 4.7.1. Cell Culture

Cells were grown in RPMI 1640 medium supplemented with 10% fetal bovine serum (FBS) and 1% penicillin/streptomycin and maintained in a humidified atmosphere with 5% CO_2_ at 37 °C. Medium changes were made every 2–3 days for maintenance. Subcultures were made twice a week until they reached approximately 80% confluence for assays.

#### 4.7.2. MTT Assay

The MTT assay is based on the ability of dehydrogenase enzymes from metabolically viable cells to reduce tetrazolium rings and form formazan crystals; consequently, the number of viable cells is directly proportional to the level of formazan produced [55,56,57]. VERO cells were seeded in 96-well microplates (Nest^®^) at a density of 4 × 10^4^ cells/cm^2^, which allowed their adhesion and proliferation for 24 h. After this time, the cells were incubated with RPMI 1640 medium containing different concentrations of ISO (9.76 to 2500 µg/mL) for 24 h. After incubation, the treatments were removed, and 100 μL of MTT was added to each well at a concentration of 0.125 mg/mL. The plate was incubated at 5% CO_2_ at 37 °C for 4 h. Subsequently, the MTT was discarded, and the formazan crystals deposited at the bottom of each well were dissolved in 100 μL of dimethylsulfoxide (DMSO). Absorbance was determined at an optical density (OD) of 570 nm using an Epoch 2 microplate reader (Biotek). Cells without treatments were used as negative controls. The absorbance values were normalized by considering the absorbance obtained from untreated cultures as 100%.

#### 4.7.3. Crystal Violet Assay (CV)

The CV assay is based on the staining of the DNA and proteins of the cells available in the culture wells, and the color intensity is proportional to the number of viable cells [45]. VERO cells were seeded in 96-well microplates at a density of 4 × 10^4^ cells/cm^2^, allowing their adhesion and proliferation for 24 h. After this time, the cells were incubated with RPMI 1640 medium containing ISO (at previously described concentrations) for 24 h. At the end of the treatments, the medium was removed, and the cells were washed with PBS, then fixed by depositing 100 μL of 4% paraformaldehyde solution in each well for 30 min at room temperature. Paraformaldehyde was discarded, and 100 μL of 0.5% CV solution in 6% methanol was added to each well for 30 min at room temperature. The CV was discarded, and each well was carefully rinsed with distilled water until the remaining dye was extracted. The plate was allowed to air dry for 24 h. To extract the CV bound to the DNA, 200 μL/well of methanol was used, and the plate was incubated for 20 min at room temperature on an orbital shaker with a frequency of 20 oscillations per minute. Finally, the absorbance was determined as indicated above. The results for both assays were expressed through dose–response curves using 16 ISO concentrations (9.76 to 2500 µg/mL). IC_50_ values (50% inhibitory concentration of the cell population) were calculated from the fit (R^2^ > 0.95) of the Hill slope curve of the experimental data using nonlinear regression analysis in GraphPad Prism version 8.0 software.

### 4.8. Data Analysis

Results were analyzed using GraphPad Prism version 8.0 software. Normality was assessed using the Shapiro–Wilk test. One-way ANOVA was performed to assess the impact of ISO treatment on the inhibition of biofilm formation compared to the untreated control group; to compare the effects of ISO and AFB on the inhibition of biofilm formation, Tukey’s test was used; to evaluate the effect of ISO on the integrity of the wall, the Tukey test was also used; and to evaluate the ΔΨm, Dunn’s test was used.

## 5. Conclusions

In this investigation, we explore other antifungal action targets of ISO. Our results show that ISO has the ability to inhibit the formation of fungal biofilms, causes the loss of ΔΨm with the consequent mitochondrial dysfunction, and can affect the cell wall of these pathogens. Confirming that the monoterpene ISO has different targets of action against these yeasts. This intensifies the interest in continuing to investigate the mechanisms of action of this compound, which could be used as an adjuvant in the treatment and control of pathogenic yeasts resistant to antifungals.

## Figures and Tables

**Figure 1 ijms-24-10187-f001:**
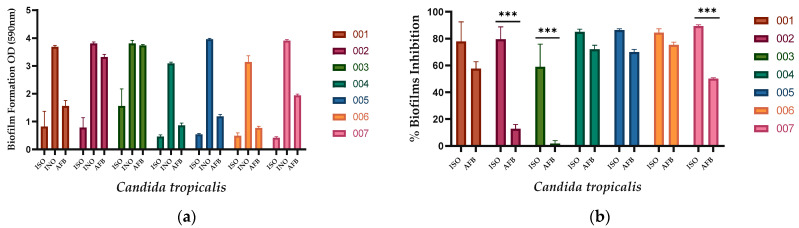
Action of ISO and AFB on the formation of biofilms. (**a**) Biofilm formation at 37 °C for 48 h. Where, ISO: cells treated with ISO; INO: untreated cells (negative control); AFB: cells treated with AFB (4 µg/mL). (**b**) Percentage reduction in biofilm formation after treatment with ISO and AFB. The ANOVA results showed a value of *** *p* < 0.001, and the Tukey test with a confidence level of 95% indicates that there are significant differences between the effect of ISO and the effect of AFB (for isolates 002, 003, and 007); while for the rest of the isolates (001, 004, 005, and 006) there are no significant differences.

**Figure 2 ijms-24-10187-f002:**
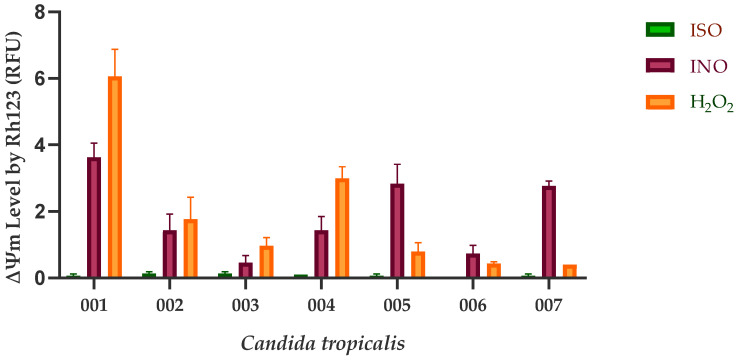
Mitochondrial depolarization of *C. tropicalis* caused by ISO. Depolarization of the mitochondrial membrane was detected by Rh123 staining, and the fluorescence intensity of the cells was analyzed using a flow cytometer. RFU, relative fluorescent units. INO corresponds to the negative control group (cells without treatment). H_2_O_2_ (15 mM) was used as a positive control for disruption of the mitochondrial membrane potential (ΔΨm). The results are expressed as the mean ± standard deviation of three independent experiments; Dunn’s test shows us that there are statistically significant differences between the ISO-INO and ISO-H_2_O_2_ treatments (*p* < 0.005); however, these differences do not exist between the INO-H_2_O_2_ treatments (*p* > 0.005).

**Figure 3 ijms-24-10187-f003:**
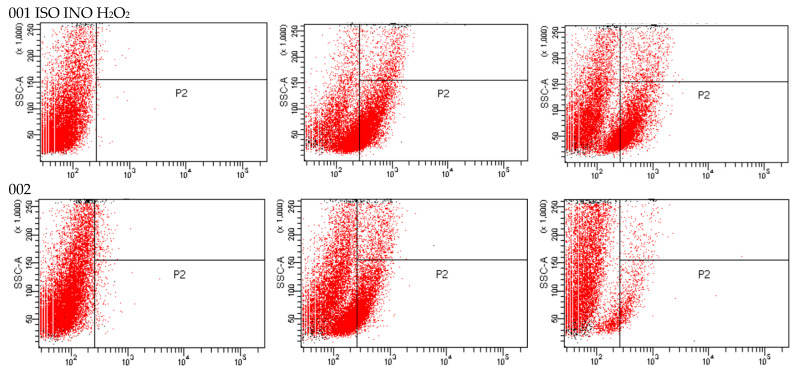
Rh123 fluorescence emitted by ISO-treated, INO-treated, and H_2_O_2_-treated *C. tropicalis* yeasts.

**Figure 4 ijms-24-10187-f004:**
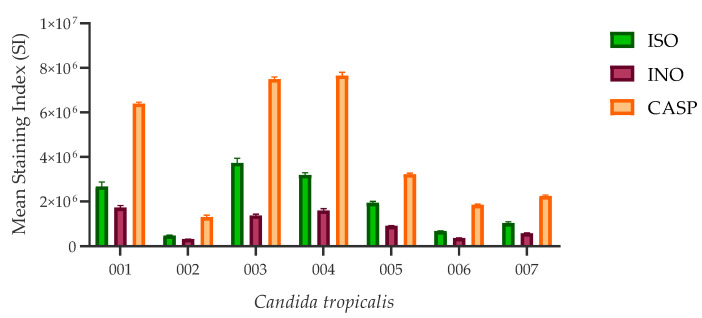
Chitin content in the cell wall of *C. tropicalis*. ISO-treated cells have higher SI values (indicating higher chitin content revealed by CFW staining) compared to untreated (INO) cells; however, these values are lower compared to CASP-treated cells, although there are no statistically significant differences between the ISO vs. INO and ISO vs. CASP treatments (*p* > 0.005).

**Figure 5 ijms-24-10187-f005:**
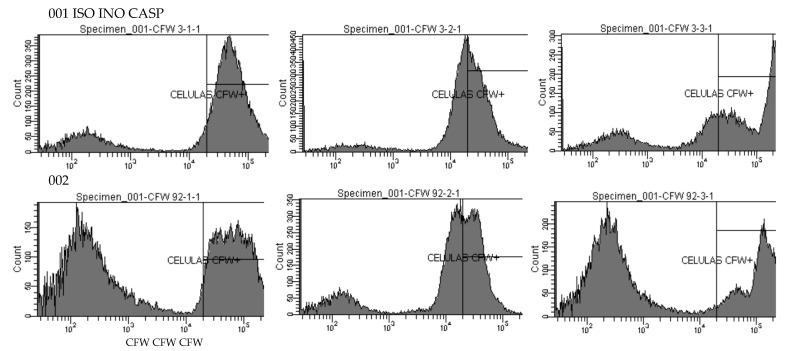
Histograms of the fluorescence emitted by isolates 001 and 002. ISO (cells treated with ISO); INO (untreated cells) and CASP (CASP-treated cells), after staining with 2.5 µg/mL CFW.

**Figure 6 ijms-24-10187-f006:**
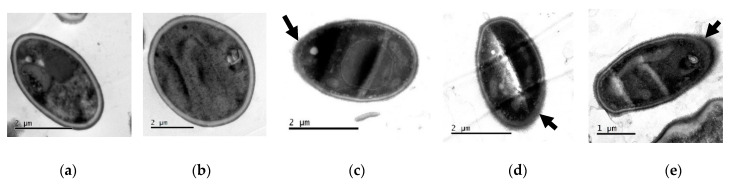
TEM of *C. tropicalis* untreated (**a**,**b**) and treated with ISO (**c**–**e**). Changes (indicated with the arrows) are evident in the morphology of the cells treated with ISO, as well as damage to the integrity of the cell wall.

**Figure 7 ijms-24-10187-f007:**
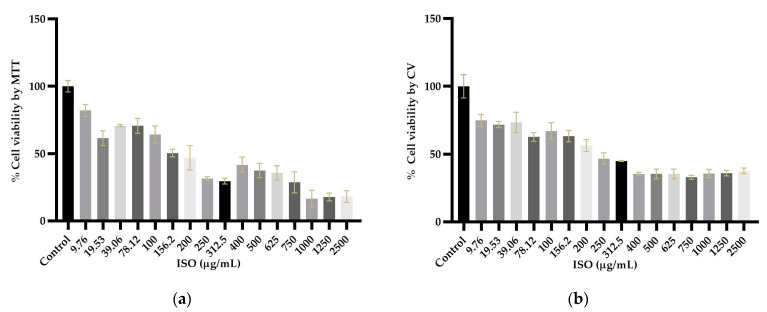
Percentage of viability of VERO cells exposed to ISO (9.76–2500 μg/mL) using the MTT (**a**) and CV (**b**) methods. With both methods, a 50% reduction in viability is observed from 250 μg/mL. Results are expressed as mean ± standard error of the mean. The percentages are expressed in relation to the control, and significant differences are evident between the cells treated with ISO (*p* < 0.05).

**Figure 8 ijms-24-10187-f008:**
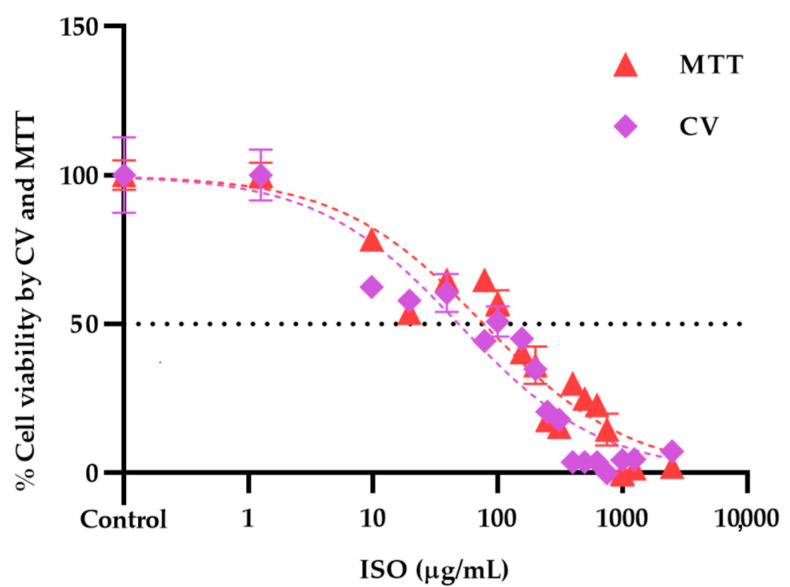
Dose–response curves of the effect of ISO (9.76–2500 μg/mL) on the viability of VERO cells through the CV and MTT assays at 24 h; the calculated IC_50_ was 48.64 and 77.34 μg/mL, respectively. Results are expressed as mean ± standard error of the mean.

**Table 1 ijms-24-10187-t001:** Minimum inhibitory concentration (MIC) and minimum fungicidal concentration (MFC) [µg/mL] of ISO against *C. tropicalis*.

*Candida tropicalis*	ISO		
	MIC_90_	MIC_50_	MFC
CLI 001	470	261.2	500
CLI 002	326.6	59.38	350
CLI 003	413.3	124.4	400
CLI 004	420.8	121.5	450
CLI 005	500	234.6	500
CLI 006	463.9	179.8	450
CLI 007	391.6	107	400

**Table 2 ijms-24-10187-t002:** Percentages of inhibition of the formation of fungal biofilms of ISO vs. AFB in *C*. *tropicalis*.

*Candida tropicalis*	ISO	AFB
001	77.80	57.70
002	79.48	12.86
003	59.18	1.87
004	85.09	72.23
005	86.46	70.09
006	84.38	75.45
007	89.35	50.30

## Data Availability

The data presented in this study are available in the article.

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
