# Peer review of "Isoespintanol Antifungal Activity Involves Mitochondrial Dysfunction, Inhibition of Biofilm Formation, and Damage to Cell Wall Integrity in Candida tropicalis"

_ijms, 2023, doi:10.3390/ijms241210187_

Round 1
Reviewer 1 Report

Manuscript needs to checked over for spelling and grammatical errors.
Reviewer 2 Report
The manuscript evaluates isoespintanol (ISO) effect on Candida tropicalis in planktonic and biofilm states, its mechanism of action against the mitochondrial membrane potential and the integrity of the cell wall. Furthermore, this study explores the possible cytotoxicity of this compound. I would recommend some minor revisions before publication.
In lines 212-219, the authors assert that ISO can eradicate C. tropicalis biofilms. I completely disagree. The assay performed in the study is only able to detect biofilm biomass, not viability. The biomass loss can be attributed to the inhibition of yeast growth or a disaggregating ability coming from the compound. The yeast biofilm eradication can be only asserted when a complete blastopores production from the biofilm is detected at the same time that a complete viability loss in the bottom-adhered biofilm. Considering the results obtained are similar to those one obtained for amphotericin B, I’ll suggest the ISO effect on biofilms is inhibitory and not eradicating.
In the Discussion section, considering the potential ISO cytotoxicity, could the author give some potential uses for this kind of compound in the clinic?
Reviewer 3 Report
The aim of the presented manuscript was to investigate the effect of isoespintanol (ISO), a monoterpene, (ISO) on Candida tropicals. The authors found that ISO inhibited biofilm formation, caused mitochondrial dysfunction, and affected cell wall integrity, leading to overproduction of chitin. These studies are a continuation of earlier studies published by the authors in [19] doi.org/10.3390/molecules27185808. My main concern is the lack of novelty in the case of some results:
1) The effect of ISO on biofilm has been already tested in the previous paper [19], although in a different experimental setup than in the manuscript.
2) Lines 78-80. The authors claim that Table 1 shows the values of MIC and MFC “evaluated in this study and in previous work [19]” but the entire dataset was copied from Table 1 in [19].
3) Fig. 6, TEM of C. tropicalis untreated and treated with ISO, shows similar results as in Figure 10c from [19].
Point 1) was discussed in the manuscript (lines 212-221), but the other two experiments seem redundant.
Minor comments
Lines 16-17 -should be “…fungal biofilms,”
Figure 2. The y-axis title should be changed to the English version.
Figure 3 and Figure 4. The titles of the X-axis - shouldn’t they be changed for rhodamine and calcofluor, respectively?
